# Learning a Neural Pareto Manifold Extractor with Constraints

Soumyajit Gupta[1]    Gurpreet Singh[2]    Raghu Bollapragada[3]    Matthew Lease[4]

[1]Department of Computer Science, University of Texas at Austin, USA
[2]XtractorAI
[3]Operations Research and Industrial Engineering, University of Texas at Austin, USA
[4]School of Information, University of Texas at Austin, USA

## Abstract

Multi-objective optimization (MOO) problems require balancing competing objectives, often under constraints. The *Pareto optimal* solution set defines all possible optimal trade-offs over such objectives. In this work, we present a novel method for *Pareto-front learning*: inducing the full Pareto manifold at train-time so users can pick any desired optimal trade-off point at run-time. Our key insight is to exploit Fritz-John Conditions for a novel guided *double gradient descent* strategy. Evaluation on synthetic benchmark problems allows us to vary MOO problem difficulty in controlled fashion and measure accuracy *vs.* known analytic solutions. We further test scalability and generalization in learning optimal neural model parameterizations for Multi-Task Learning (MTL) on image classification. Results show consistent improvement in accuracy and efficiency over prior MTL methods as well as techniques from operations research.

## 1 INTRODUCTION

Multi-Objective Optimization (MOO) problems require balancing multiple objectives, often competing with one another under further constraints [Van Rooyen et al., 1994, Ehrgott and Wiecek, 2005]. A *Pareto optimal* solution [Pareto, 1906] defines the set of all saddle points [Ehrgott and Wiecek, 2005] such that no objective can be further improved without penalizing at least one other objective.

As operational systems today increasingly seek to balance competing objectives, research on Pareto optimal learning has quickly grown across tasks such as fair classification [Balashankar et al., 2019, Martinez et al., 2020], diversified ranking [Liu et al., 2019, Sacharidis, 2019], and recommendation [Xiao et al., 2017b, Azadjalal et al., 2017]. Many practical classification and recommendation problems have been shown to be non-convex [Hsieh et al., 2015]. A general Pareto solver should thus support optimization for both non-convex objectives and constraints.

Because MOO problems typically lack a single global optimum, one must choose among optimal solutions by selecting a trade-off over competing objectives. Ideally this choice could be deferred to run-time, so that each user could choose whichever trade-off they prefer. Unfortunately, prior Pareto solvers have typically required training a separate model to find the Pareto solution point for each desired trade-off.

To address this, recent work has proposed *Pareto front learning* (PFL): inducing the full Pareto manifold in training so that users can quickly select any desired optimal trade-off point at run-time [Navon et al., 2021, Lin et al., 2020, Singh et al., 2021]. These works learn a neural model manifold to map any desired trade-off over objectives to a corresponding Pareto point. See **Appendix L** for additional motivation for PFL. As with other supervised learning, inducing an accurate prediction model requires high quality training data, *i.e.,* Pareto points used for training should be accurate.

In this work, we devise a efficient Pareto search procedure for Singh et al. [2021]'s HNPF model, so that we may benefit from its correctness guarantees in identifying true Pareto points for PFL training. While HNPF supports non-convex MOO with constraints and bounded error, it suffers from a lack of scalability with increasing variable space. Our innovation is a novel, guided *double gradient descent* strategy, updating the candidate point set in the outer descent loop and the manifold estimators in the inner descent loop.

Our evaluation spans both synthetic benchmarks and multi-task learning (MTL) problems. Benchmark problems allow us to conduct controlled experiments varying MOO problem complexity (*e.g.,* the presence of constraints and/or convexity in variable or function domains). Analytic solutions to benchmark problems enable us to measure the true accuracy of model predictions, something which is often difficult or impossible on real-world problems. Additional evaluation on a set of MTL problems in image classification enable us to further test scalability and generalization in learning Pareto optimal weights for high dimensional neural models.

Results across synthetic benchmarks and MTL problems

*Accepted for the 38th Conference on Uncertainty in Artificial Intelligence* (UAI 2022).

show clear, consistent advantages of SUHNPF in terms of capability (handling non-convexity and constraints), denser coverage and higher accuracy in recovering the true Pareto front, and greater efficiency (time and space). Beyond empirical findings, our conceptual framing and review of prior work also serves to further bridge complementary lines of prior work in MTL and operations research. For reproducibility, we share our sourcecode and data[1].

## 2 DEFINITIONS

We adopt Pareto definitions from Marler and Arora [2004]. A general MOO problem can formulated as follows:

$$\text{optimize} \quad F(x) = (f_1(x), \ldots, f_k(x)) \quad (1)$$

$$\text{s.t.} \quad x \in S = \{x \in \mathbb{R}^n | G(x) = (g_1(x), \ldots, g_m(x)) \le 0\}$$

with $n$ variables $(x_1, \ldots, x_n)$, $k$ objectives $(f_1, \ldots, f_k)$, and $m$ constraints $(g_1, \ldots, g_m)$. Here, $S$ is the feasible set, *i.e.,* the set of input values $x$ that satisfy the constraints $G(x)$. For a MOO problem optimizing $F(x)$ subject to $G(x)$, the solution is usually a manifold as opposed to a single global optimum, therefore one must find the set of all points that satisfy the chosen definition for an optimum.

**Strong Pareto Optimal:** A point $\tilde{x}^* \in S$ is *strong* Pareto optimal if no point in the feasible set exists that improves an objective without detriment to at least one other objective.

$$\nexists x_j : f_p(x_j) \le f_p(x^*), \quad \text{for} \quad p = 1, 2, \ldots, k$$
$$\exists l : f_l(x_j) < f_l(x^*) \quad (2)$$

**Weak Pareto Optimal:** A point $\tilde{x}^* \in S$ is *weak* Pareto optimal if no other point exists in the feasible set that improves all of the objectives simultaneously. This is different from strong Pareto, where points might exist that improve at least one objective without detriment to another.

$$\nexists x_j : f_p(x_j) < f_p(\tilde{x}^*), \quad \text{for} \quad p = 1, 2, \ldots, k \quad (3)$$

## 3 RELATED WORK

**Linear Scalarization (LS).** A variety of work has adopted LS to find Pareto points [Xiao et al., 2017b, Lin et al., 2019, Milojkovic et al., 2019]. LS converts an MOO into a SOO using a convex combination of objective functions and constraints. However, because Karush-Kuhn-Tucker (KKT) conditions are known to hold true only for convex cases [Boyd et al., 2004], LS solutions are guaranteed to be Pareto optimal only under fully convex setting of objectives and constraints, as shown in Gobbi et al. [2015].

**Operations Research (OR).** A variety of OR methods support MOO problems with non-convex objectives and constraints, guaranteeing correctness within a user-specified error tolerance. Correctness has also been further verified by evaluation on synthetic MOO benchmark problems with known, analytic solutions. However, a key limitation of these methods is lack of scalability: they suffer from significant computational and run-time limitations as the variable

[1] https://github.com/smjtgupta/SUHNPF

dimension increases. Hence, they cannot be applied to optimizing neural model parameters for MOO problems.

Table 1: SUHNPF *vs.* existing Operations Research (OR) and Multi-Task Learning (MTL) methods. OR methods account for both objectives and constraints, produce Pareto points only, and are known to find true Pareto points for non-convex MOO problems. However, these methods do not scale to high-dimensional neural MOO problems. In contrast, MTL methods scale well but typically do not support constraints and can struggle with non-convexity.

| Type | Method | Finds Only Pareto points | Handles Constraints | Scalable Neural MOO |
|---|---|---|---|---|
| Operations Research (OR) | NBI [1998] | ✓ | ✓ | ✗ |
| | mCHIM [2015] | ✓ | ✓ | ✗ |
| | PK [2016] | ✓ | ✓ | ✗ |
| | HNPF [2021] | ✓ | ✓ | ✗ |
| Multi-Task Learning (MTL) | MOOMTL [2018] | ✗ | ✗ | ✓ |
| | PMTL [2019] | ✗ | ✗ | ✓ |
| | EPO [2020] | ✗ | ✗ | ✓ |
| | EPSE [2020] | ✗ | ✗ | ✓ |
| | PHN [2021] | ✗ | ✗ | ✓ |
| Ours | **SUHNPF** | ✓ | ✓ | ✓ |

Examples include enhanced scalarization approaches such as NBI [Das and Dennis, 1998], mCHIM [Ghane-Kanafi and Khorram, 2015], and PK [Pirouz and Khorram, 2016]. NBI produces an evenly distributed set of Pareto points given an evenly distributed set of weights, using the concept of Convex Hull of Individual Minima (CHIM) to break down the boundary/hull into evenly spaced segments before tracing the *weak* Pareto points. mCHIM improves upon NBI via a quasi-normal procedure to update the aforementioned CHIM set iteratively, to obtain a strong Pareto set. PK uses a local $\epsilon$-scalarization based strategy that searches for the Pareto front using controllable step-lengths in a restricted search region, thereby accounting for non-convexity.

**Multi-Task Learning (MTL).** Recent MTL works have devised Pareto solvers for estimating high-dimensional neural models. MOOMTL [Sener and Koltun, 2018] effectively scales via a multi-gradient descent approach, but does not guarantee an even spread of solution points found along the Pareto front. PMTL [Lin et al., 2019] addresses this spread issue by dividing the functional domain into equal spaced cones, but this increases computational complexity as the number of cones increases. EPO [Mahapatra and Rajan, 2020] extends preference rays along specified weights to find Pareto points evenly spread in the vicinity of the rays. EPSE [Ma et al., 2020] uses a combination Hessian of the functions and Krylov subspace to find Pareto solutions.

MTL methods rely upon KKT conditions to check for optimality, which assumes convexity (see earlier LS discussion). While methods seek an even distribution of Pareto points by dividing the functional space into evenly spaced cones or preference rays, our results on a non-convex benchmark problem clearly show an uneven point spread (**Section 6.1**). Moreover, most MTL methods are *point-based solvers*, meaning they must be run $P$ times to find $P$ points. This is too expensive to adjust trade-off preferences at run-time.

**Pareto front learning**. PFL methods [Navon et al., 2021, Lin et al., 2020, Singh et al., 2021] induce the full Pareto manifold at train-time so that users can quickly select any desired optimal trade-off point at run-time. For example, a manifold model trained on $P$ Pareto points might then quickly produce any number of additional Pareto points via interpolation. Of course, quality training data quality is necessary to learn an accurate, supervised prediction model. The method and resulting accuracy of the Pareto points used for model training is thus crucial to prediction accuracy.

Navon et al. [2021]'s PHN considers two way to acquire Pareto training points: LS and EPO [2020]. Lin et al. [2020] use their PMTL [2019] method to identify Pareto points for training. Singh et al. [2021]'s HNPF uses the Fritz-John conditions (FJC) [Maruşciac, 1982] to identify Pareto points.

Like other OR methods, HNPF provides a theoretical guarantee of Pareto front accuracy within a user-specified error tolerance. In evaluation on canonical OR benchmark problems, HNPF was shown to recover known Pareto fronts across various non-convex MOO problems while also being more efficient in finding Pareto points than NBI [1998], mCHIM [2015], and PK [2016]). However, like other OR methods, HNPF cannot scale to learn optimal high-dimensional neural model weights for MOO problems.

Ha et al. [2017]'s hypernetworks proposed training one neural model to generate effective weights for a second, target model. Navon et al. [2021] and Lin et al. [2020] apply this approach to learn a manifold mapping MOO solutions to different target model weights, enabling the target model to achieve the desired Pareto trade-off for the MOO problem. However, HNPF cannot be similarly applied to MTL problems due to its lack of scalability.

## 4 PRELIMINARIES

**Fritz John Conditions (FJC)**. Let the objective and constraint function in Eq. (1) be differentiable once at a decision vector $x^* \in \mathcal{S}$. The Fritz-John [Levi and Gobbi, 2006] necessary conditions for $x^*$ to be *weak* Pareto optimal is that vectors must exists for $0 \leq \lambda \in \mathbb{R}^k, 0 \leq \mu \in \mathbb{R}^m$ and $(\lambda, \mu) \neq (0, 0)$ (not identically zero) *s.t.* the following holds:

$$\sum_{i=1}^{k} \lambda_i \nabla f_i(x^*) + \sum_{j=1}^{m} \mu_j \nabla g_j(x^*) = 0 \qquad (4)$$

$$\mu_j g_j(x^*) = 0, \forall j = 1, \ldots, m$$

Gobbi et al. [2015] present an $L$ matrix form of FJC:

$$L = \begin{bmatrix} \nabla F & \nabla G \\ \mathbf{0} & G \end{bmatrix} \quad [(n+m) \times (k+m)] \qquad (5)$$

$$\nabla F_{n \times k} = [\nabla f_1, \ldots, \nabla f_k]$$
$$\nabla G_{n \times m} = [\nabla g_1, \ldots, \nabla g_m]$$
$$G_{m \times m} = diag(g_1, \ldots, g_m)$$

comprising the gradients of the functions and constraints. The matrix equivalent of FJC for $x^*$ to be Pareto optimal

is to show the existence of $\delta = (\lambda, \mu) \in \mathbb{R}^{k+m}$ (*i.e.*, $\delta$ not identically zero) in Eq. (4) such that:

$$L \cdot \delta = 0 \quad \text{s.t.} \quad L = L(x^*), \delta \geq 0, \delta \neq 0 \qquad (6)$$

Therefore the non-trivial solution for Eq. (6) is:

$$det(L^T L) = 0 \qquad (7)$$

**Remark.** *If $f_i$s and $g_j$s are continuous and differentiable once, then the set of weak Pareto optimal points are $x^* = \{x | det(L(x)^T L(x)) = 0\}$, $\delta \geq 0$ for a non-square matrix $L(x)$, and is equivalent to $x^* = \{x | det(L(x)) = 0\}$, $\delta \geq 0$, for a square matrix $L(x)$.* See **Appendix C** for a proof of the above for the unconstrained setting only.

**Hybrid Neural Pareto Front (HNPF)**. Like other Pareto front learning (PFL) methods, HNPF [Singh et al., 2021] learns a neural Pareto manifold from training data. With HNPF, Pareto points are acquired from training data via Fritz-John conditions. In particular, once a given a data point from the input variable domain is mapped to the output function domain (via objective functions), FJC are tested to determine Pareto optimality for that point.

HNPF's neural network first identifies *weak Pareto* points via feed-forward layers to smoothly approximate the *weak* Pareto optimal solution manifold $M(X^*)$ as $\tilde{M}(\tilde{X}, \Phi)$. The last layer of the network has two neurons with *softmax* activation for binary classification of Pareto *vs.* non-Pareto points, distinguishing sub-optimal points from the *weak* Pareto points. The network loss is representation driven, since the Fritz John discriminator (Eq. (7)), described by the objective functions and constraints, explicitly classifies each input data point $X_i$ as being *weak* Pareto or not.

After identifying weak Pareto points, HNPF uses an efficient Pareto filter to find the subset of *non-dominated* points.

HNPF's scalability bottleneck lies in how it samples variable domain points to test for Pareto optimality in model training. If there are any direct constraints on variable values, this naturally restricts the feasible domain for sampling. However, lacking any prior distribution on where to find Pareto optima, HNPF performs uniform random sampling in the variable domain to ensure broad coverage for locating optima. For small benchmark problems with known variable domains, this suffices. However, it is infeasible to apply this to find optimal model parameters for a neural MOO model.

## 5 SCALABLE UNIDIRECTIONAL HNPF

To address HNPF's scalability bottleneck, we introduce SUHNPF, a scalable variant of HNPF for finding weak Pareto points with an arbitrary density and distribution of initial data points. This is achieved via a scalable unidirectional FJC-guided double-gradient descent algorithm that encompasses HNPF's neural manifold estimator. Given continuous differentiable loss functions, SUNHPF's guided double gradient descent strategy efficiently searches the variable domain to find Pareto optimal points in the function domain. This enables SUHNPF to learn an $\epsilon$-bounded approximation $\tilde{M}(\Theta^*)$ to the weak Pareto optimal manifold.

## 5.1 FJC-GUIDED DOUBLE GRADIENT DESCENT

Constructing a classification manifold of Pareto *vs.* non-Pareto points requires a set of feasible points to represent both classes. Since the Pareto manifold is unknown *a priori*, feasible points are drawn from a random distribution (lacking an informed prior) to initialize both classes. We then refine the points in the Pareto class $\mathcal{P}1$ while holding the non-Pareto points $\mathcal{P}0$ constant.

We assume an equal-sized sample set of $P$ points for each class, which helps to address class imbalance for harsh cases. For benchmark problems where the feasible set over the variable domain is known, we randomly sample points over this feasible domain to initialize $\mathcal{P}1$ and $\mathcal{P}0$. Given these input points $x$, held constant for $\mathcal{P}0$ and used as initial seed values for $\mathcal{P}1$, **Alg. 1** specifies our FJC-guided double-gradient descent algorithm. The algorithm iteratively updates $\mathcal{P}1$ towards the Pareto manifold via FJC-guided descent. The training dataset $D$ is the union of $\mathcal{P}0 \cup \mathcal{P}1$. The algorithm iterates over Steps 5-9 until the error $(err)$ converges to the user-specified error tolerance $(\epsilon_{outer})$.

$$err = \sum_{p \in \mathcal{P}1} \left( det(L^T L) \right)^2 \tag{8}$$

---
**Algorithm 1** FJC-guided descent of variable domain
---
1: **Input**: Data $D = \mathcal{P}0 \cup \mathcal{P}1$      ▷ Training Data
2: **Input**: Functions $F$ and Constraints $G$
3: **Input**: Error tolerance $\epsilon_{outer}, \epsilon_{inner}$
4: **while** $err > \epsilon_{outer}$ **do**      ▷ Run until convergence
5:      Train network using $D$ as data for $e$ epochs
6:      Compute current error $err$
7:      Compute $\nabla_p det = \frac{\partial det(L^T L)}{\partial p}, \forall p \in \mathcal{P}1$
8:      $\mathcal{P}1 \leftarrow \mathcal{P}1 - \eta \nabla det$      ▷ Update points in $\mathcal{P}1$
9:      $D = \mathcal{P}0 \cup \mathcal{P}1$      ▷ Update Training Data
10: **Output**: Weak Pareto manifold $\tilde{M}$
---

Eq. 8 in Alg. 1 ensures that all of the points in the Pareto set ($p \in \mathcal{P}1$) are optimal once we converge to the desired error tolerance $\epsilon$. Hence, Step 7 computes gradients of the $det(L^T L)$ matrix *w.r.t.* the variables at points $p \in \mathcal{P}1$ and creates an approximation of the $\nabla det$ matrix. The training data $D$ is then updated with the new values of $\mathcal{P}1$. The output is an approximation of the true weak Pareto manifold $M$ as $\tilde{M}$ on the discrete dataset $D \subset X$. Note that in Step 8, we do not allow the point set $\mathcal{P}1$ to leave the feasible set $S$ *i.e.*, if the step crosses the boundary of the feasible set, then we update the point to be the point on the boundary.

Alg. 1 includes two separate gradient descent steps. The outer descent loop (Step 4-9) updates the candidate point set $\mathcal{P}1$ using the error measurement of $err$ through a squared loss in Eq. 8. The inner descent (Step 5) updates the parameters ($\Phi$) of the neural net to closely approximate the Pareto manifold $M(X)$ as $\tilde{M}(X, \Phi)$. This is done using the Binary Cross Entropy Loss on $(det(L(X)^T L(X)), \tilde{M}(X))$, and reaches convergence only when $BCE \leq \epsilon_{inner}$. The

*unidirectional* property of this double-gradient update lets the outer loop influence the inner loop but not vice-versa.

**Complexity Analysis**. The time complexity of the proposed Alg. 1 is $\mathcal{O}(\mathcal{P}(k+m)^2 n + \mathcal{P}(k+m)^3)$. Under a practical deep MTL, $n \gg k, m$ (*i.e.*, variable dimension is strictly greater than the number of functions and constraints in any neural setting), the complexity is dominated by the term $\mathcal{O}(\mathcal{P}(k+m)^2 n)$, where the scaling is linear in terms of the variable dimension $n$, and quadratic in the number of functions and constraints $k, m$. The space complexity is $O(n(k+m+P)) + (k+m)^2$. SUHNPF achieves better memory and run-time efficiency since it does not rely upon solving primal and dual problems used in MTL methods, with detailed analysis in **Appendix I** and **Appendix J**.

## 6 BENCHMARKING

**Motivation.** Lack of analytical solutions to real MOO problems makes it difficult to measure the true accuracy of any Pareto solver. Consequently, we follow the OR literature in advocating that the correctness of any proposed Pareto solver should first be tested on constructed benchmark problems with known analytic solutions. This is also consistent with broader ML community practice of first evaluating proposed methods across a range of simulated, controlled conditions to verify correctness, often yielding valuable insights into model behavior prior to evaluation on real data.

We consider three such benchmark problems (Cases I-III). These problems are non-convex in either the functional or variable domain, or due to constraints (**Table 2**). Note that *whether or not the Pareto front itself is non-convex is not always the best indicator of benchmark difficulty*. For example, even though both objectives are non-convex in Case II, the Pareto front is still convex. As we shall see, PHN [Navon et al., 2021] fails on Case II despite performing well on two benchmark problems in their own study having a non-convex front. In general, non-convexity can greatly challenge MTL approaches relying on KKT conditions in testing solutions for optimality (see **Appendix E**).

Table 2: Characterization of benchmark cases, including convexity (C) *vs.* non-convexity (NC) in variable and function domains.

| Case | Dim | Variable Domain | Function Domain | Includes Constraints | OR Methods | MTL Methods | SUHNPF |
|------|-----|-----------------|-----------------|---------------------|------------|-------------|--------|
| I | 2 | Linear | NC | No | Sparse, Slow | Sparse, Fast | Dense, Fast |
| II | 30 | NC | C | No | Sparse, Slow | Fail | Dense, Fast |
| III | 2 | NC | NC | Yes | Sparse, Slow | Fail | Dense, Fast |

**Experimental Setup**. For each Case I-III, each method is tasked with finding $P = 50$ Pareto points. OR methods search until any $P$ Pareto points are found. MTL methods divide the functional search quadrant into cones/rays, seeking one Pareto point per split. Manifold-based methods (PHN, HNPF, and SUHNPF) search for $P$ Pareto points in order to learn the manifold. Ideally, each method should identify an even spread (*i.e.*, broad coverage) of points across the true Pareto front (shown in grey in each figure) in order to

faithfully approximate it. We report the runtime taken by each solver to find the points.

SUHNPF starts with $P$ random candidates that are progressively refined via its guided, double gradient descent strategy. Following HNPF [Singh et al., 2021], we adopt the same error tolerance $10^{-4}$ for both $\epsilon_{outer}$ and $\epsilon_{inner}$. Any point $x$ that satisfies $|det(L(x)^T L(x))| \leq \epsilon_{inner}$ is thus classified as being Pareto (exact zero is often impossible given finite machine precision). Sourcecode for LS, MOOMTL, PMTL and EPO solvers are taken from EPO's repository, while EPSE and PHN's sourcecode are used for them, respectively (see **Appendix D**). Based on Navon et al. [2021]'s findings, we evaluate the more accurate PHN variant, PHN-EPO, which we refer to simply as PHN.

Due to key differences between OR *vs.* MTL methods, results for each group are presented separately. First, OR methods not only support the full range of non-convex conditions across Cases I-III, but provide error tolerance parameters to guarantee correctness (and our experiments confirm this). Consequently, we report only the efficiency of OR methods in Table 3. In contrast, MTL methods produced variable accuracy on Case I and failed entirely on Cases II and III (as shall be discussed). Consequently, Table 4 reports accuracy and efficiency of MTL methods for Case I only.

**Appendix D** discusses experimental setup, **Appendix G** has two other benchmarks, and **Appendix H** has loss profiles.

### 6.1 CASE I: Ghane-Kanafi and Khorram [2015]

$$f_1(x_1, x_2) = x_1, \; f_2(x_1, x_2) = 1 + x_2^2 - x_1 - 0.1sin3\pi x_1$$
$$\text{s.t.} \quad g_1 : 0 \leq x_1 \leq 1, g_2 : -2 \leq x_2 \leq 2$$

The analytical Pareto solution to this joint minimization problem is $M : 0 \leq x_1 \leq 1, x_2 = 0$. In **Fig. 1** we observe SUHNPF's randomly generated point set $\mathcal{P}1$ (red dots) converges towards the true manifold $M$ as a discrete approximation $\tilde{M}$. Point set $\mathcal{P}0$ (blue dots) is held constant and serves as representatives for the (background) non-Pareto class. Iteration 5 is the last because the error falls below the user-specified $\epsilon$. The final cardinality of the weak Pareto set $|\mathcal{P}1| = P$ and any $\mathcal{P}0$ point that happens to fall within the $\epsilon_{outer}$ threshold. Hence Alg. 1 ensures 100% Pareto point density in $\mathcal{P}1$, a vast improvement from HNPF [Singh et al., 2021], where only $\approx 2\%$ density was achieved. **Fig. 2** shows functional domain convergence. SUHNPF achieves an even spread of points in the non-convex portion of the front.

**Fig. 3** presents results for Linear Scalarization (LS) and several MTL methods: MOOMTL, PMTL, EPO, EPSE, and PHN. Refer to **Appendix F** for iterative convergence plots for Case I, and **Appendix K** for evaluation measures on *uniformity* and *coverage* for the compared methods. LS successfully produces a number of points in the non-convex portions of the front, despite prior studies often asserting that LS cannot handle any non-convexity. Refer to **Appendix M** for analysis and justification.

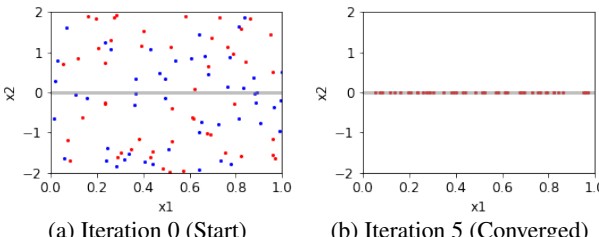

(a) Iteration 0 (Start)      (b) Iteration 5 (Converged)

Figure 1: Case I: Variable domain. The gray line show the true analytic solution ($0 \leq x_1 \leq 1$). SUHNPF Pareto candidates $\mathcal{P}1$ (red dots) converge in 5 iterations. Non-Pareto candidates $\mathcal{P}0$ (blue dots) are held constant throughout the iterative sequence.

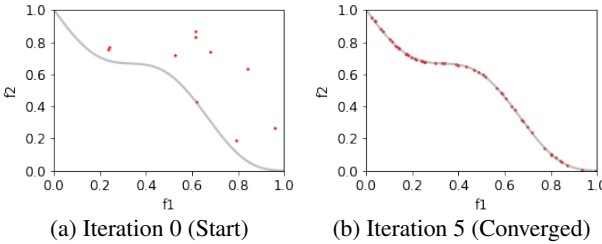

(a) Iteration 0 (Start)      (b) Iteration 5 (Converged)

Figure 2: Case I: Functional domain corresponding to Figure 1. SUHNPF Pareto candidates $\mathcal{P}1$ (red dots) converge in 5 iterations.

To check for optimality, MTL methods rely upon KKT conditions that implicitly assume convexity (see Section 3). The non-convex nature of $f_2$ is thus challenging for these KKT-based methods. For example, some methods seek an even distribution of Pareto points by breaking up the functional space into evenly spaced cones or preference rays for trade-off values $\alpha$. However, the uneven point spread seen on this non-convex benchmark illustrates limitations of the cone-based approach in handling non-convexity. We also clearly see non-Pareto points produced by some methods.

### 6.2 CASE II: Zhang et al. [2008]

$$f_1(x) = x_1 + \frac{2}{|J_1|} \sum_{j \in J_1} y_j^2 \quad , \quad f_2(x) = 1 - \sqrt{x_1} + \frac{2}{|J_2|} \sum_{j \in J_2} y_j^2$$

s.t. $g_1, \ldots, g_{30} : 0 \leq x_1 \leq 1, -1 \leq x_j \leq 1, j = 2, \ldots, m$
$J_1 = \{j | j \text{ is odd}, 2 \leq j \leq m\}, J_2 = \{j | j \text{ is even}, 2 \leq j \leq m\}$
$$y_j = \begin{cases} x_j - [0.3x_1^2 \cos(24\pi x_1 + \frac{4j\pi}{m}) + 0.6x_1] cos(6\pi x_1 + \frac{j\pi}{m}) & j \in J_1 \\ x_j - [0.3x_1^2 \cos(24\pi x_1 + \frac{4j\pi}{m}) + 0.6x_1] cos(6\pi x_1 + \frac{j\pi}{m}) & j \in J_2 \end{cases}$$

This joint minimization case operates in a $n = 30$ dimensional variable space. **Fig. 4** shows the true Pareto front and SUHNPF convergence in the variable domain. Note the non-convexity in the variable domain, where $x_1$ varies uniformly between $[0, 1]$, while $x_2, \ldots, x_{30}$ are sinusoidal in nature guided by $x_1$. Thus, the Pareto manifold has a spiral trajectory along $x_2, \ldots, x_{30}$ with evolution along $x_1$.

Despite the Pareto front being convex, the objectives are non-convex. For MTL methods, the `min_norm_solver` [Sener and Koltun, 2018], which is integral to all MTL solvers, simply fails. Consequently, no MTL results are reported.

For SUHNPF, following random initialization (iteration 0) in Fig. 4 (a), we observe that the candidate set $\mathcal{P}1$ propagates more towards increasing values of $x_1$ in Fig. 4, and approximates the expected Pareto manifold at iteration 5.

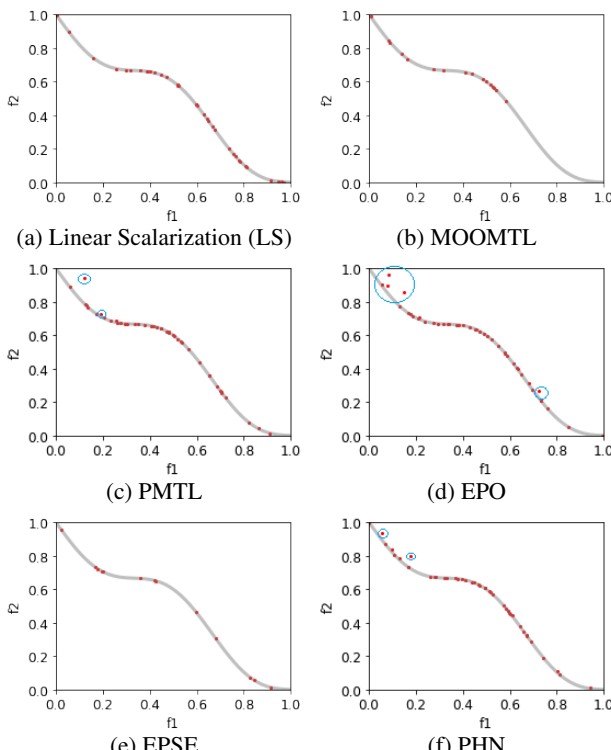

Figure 3: Case I: function domain for LS and MTL methods. No method produces all 50 of the requested Pareto points. PMTL, EPO and PHN also find non-Pareto points (circled in blue). Methods vary greatly in their coverage of points spanning the true front.

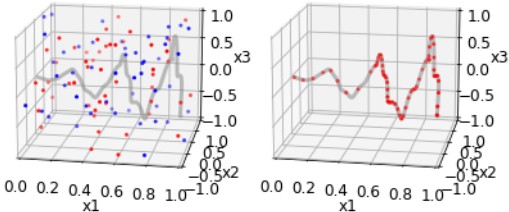

(a) Iteration 0 (Start)    (b) Iteration 5 (Converged)

Figure 4: Case II: variable domain (SUHNPF). We restrict the four plots to three dimensions ($x_1$, $x_2$, and $x_3$) for visualization.

## 6.3   CASE III: Tanaka et al. [1995]

$f_1(x_1, x_2) = x_1$, $f_2(x_1, x_2) = x_2$

s.t.   $g_1(x_1, x_2) = (x_1 - 0.5)^2 + (x_2 - 0.5)^2 \leq 0.5$

$g_2(x_1, x_2) = x_1^2 + x_2^2 - 1 - 0.1\cos(16\arctan(x_1/x_2)) \geq 0$

$g_3, g_4 : 0 \leq x_1, x_2 \leq \pi$

For this joint minimization problem, the Pareto front is dominated by the two constraints $g_1$ and $g_2$, while linear functions $f_1$ and $f_2$ do not contribute to the Pareto optimal solution. **Fig. 6** shows the convergence of SUHNPF Pareto candidates toward the known solution manifold.

Because MTL approaches do not support constraints, they are not capable of solving this benchmark problem. However, note that if we were to remove constraints $g_1$ and $g_2$, $f_1$ and $f_2$ would then become independent of each other (and so not compete). The front then collapses to the point $(0, 0)$,

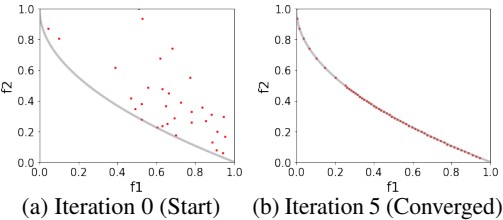

(a) Iteration 0 (Start)    (b) Iteration 5 (Converged)
Figure 5: Case II: functional domain (SUHNPF).

corresponding to the minimum of both functions. For this unconstrained problem, MTL methods would be expected to find this correct Pareto optimal solution point.

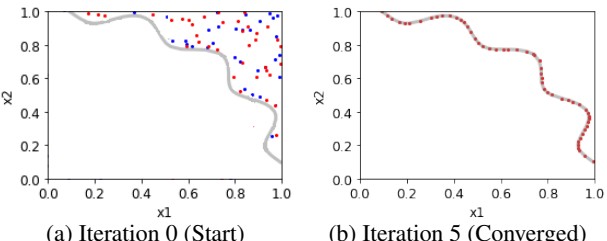

(a) Iteration 0 (Start)    (b) Iteration 5 (Converged)
Figure 6: Case III: variable domain. The analytical solution for this problem is driven by constraints $g_1, g_2$. SUHNPF Pareto candidates $\mathcal{P}0$ (red dots) converge to the true front.

Case III highlights the need for any manifold based extractor to support both explicit and implicit forms of the Pareto front. Cases I and II have explicit form of front in the functional and variable domain. However, Case III has an implicit Pareto front (Fig. 6) owing to constraints $g_1, g_2$, where they render an implicit relation between $x_1, x_2$ and therefore $f_1, f_2$. SUHNPF's ability to construct a full rank diffusive indicator function of Pareto vs. non-Pareto points enables it to approximate the true manifold.

## 6.4   SUHNPF VS. OR AND MTL METHODS

**Table 3** reports the runtime of OR methods vs. SUHNPF to find $P = 50$ Pareto points for Cases I-III. Because OR methods and SUHNPF all return $P$ true Pareto points, we compare methods on efficiency only.

Cases I and III have a 2D ($n = 2$) variable domain, where SUHNPF takes 1s per epoch, with 2 epochs for training in Step 7 of Alg. 1. Both the cases took 5 epochs to converge, resulting in a total run-time of 10s. Case II has a 30D ($n = 30$) variable domain where SUHNPF takes 2s per epoch resulting in a total run-time of 20s.

Note that HNPF [Singh et al., 2021] was shown to scale better with variable dimension $n$ in comparison to prior OR methods (e.g., see HNPF's Figure 9). Table 3 appears consistent with this: HNPF takes longer than OR baselines for variable dimension $n = 2$ (Cases I and III) but is much faster with variable dimension $n = 30$ (Case II).

Table 3: Runtime (secs) for SUHNPF vs. OR methods.

| Method | $n$ | NBI | mCHIM | PK | HNPF | **SUHNPF** |
|---|---|---|---|---|---|---|
| Case I | 2 | 14 | 13 | 13 | 45 | 10 |
| Case II | 30 | 243,344 | 67,610 | 46,808 | 3,960 | 20 |
| Case III | 2 | 36 | 41 | 37 | 75 | 10 |

**Table 4** reports the accuracy, efficiency and run-time of SUHNPF *vs.* MTL methods for Case I. For Case II, the `min_norm_solver` [Sener and Koltun, 2018] used by MTL methods fails, and Case III's constraints are not supported by MTL methods. Note that for fair evaluation, we only consider candidates that are produced within the feasible functional bounds for the problem. Additional run-time evaluation and discussion can be found in **Appendix I**.

Table 4: SUHNPF *vs.* MTL methods on Case I in finding $P = 50$ Pareto points. We report the % of feasible points each method finds and their avg/max error *vs.* the true front. Our error measure considers feasible points only; infeasible points are not penalized.

| Method | LS | MOOMTL | PMTL | EPO | EPSE | PHN | SUHNPF |
|---|---|---|---|---|---|---|---|
| Run-time (secs) | 18.1 | 19.2 | 527 | 752 | 641 | 853 | 10.0 |
| Points Found | 54% | 32% | 70% | 68% | 30% | 80% | 100% |
| Avg Err $(10^{-4})$ | 0.53 | 0.45 | 4.15 | 8.73 | 0.61 | 3.04 | 0.52 |
| Max Err $(10^{-4})$ | 1.12 | 0.98 | 126 | 106 | 0.94 | 73.8 | 0.82 |

Regarding Case I coverage and accuracy, SUHNPF returns all 50 Pareto points; no MTL method does. For all points that are found, we measure their error *vs.* the true Pareto front. SUHNPF is seen to achieve the lowest error, with maximum error bounded by the $10^{-4}$ error tolerance parameter set in our experiments. Specifically, the outer loop of Alg. 1 would not achieve convergence until all the points points are within the prescribed error tolerance. In contrast, PMTL, EPO, and PHN yield maximum error two orders of magnitude larger. Note also that our error metric generously scores only the points found by each method, with no penalty for missing points. Visually, SUHNPF (Fig. 2) clearly provides better coverage of the Pareto front via a denser, more even spread of points *vs.* those found by MTL methods (Fig. 3).

Because MTL approaches assume convexity of objective functions to generate points with uniformity on the Pareto front, and Case I includes non-convex objectives, the MTL solvers fail to find points in certain regions (see Fig. 3). While EPO's solver has convergence criteria, it still produces points that did not converge (circled in blue). This stems from EPO's assumption on KKT conditions to achieve optimality, which fails on Case I's non-convex form of $f_2$. Correspondingly PHN(-EPO), which uses EPO as its base solver, also fails to converge on certain points. In contrast, SUHNPF relies on the FJC to test optimality, which fully supports non-convexity in functions and constraints.

Regarding Case I efficiency, SUHNPF is also fastest: nearly twice as fast as LS and MOOMTL, more than 50x faster than PMTL and EPSE, 75x faster than EPO, and 85x faster than PHN. (Because PHN-EPO calls EPO, it is necessarily slower than EPO). As Navon et al. [2021] note, LS is much faster than EPO, so one could expect PHN-LS to be faster than PHN-EPO and slower than LS.

# 7 SUHNPF AS A HYPERNETWORK

Hypernetworks [Ha et al., 2017] train one neural model to generate effective weights for a second, target model. Navon et al. [2021] and Lin et al. [2020] learn a neural manifold mapping MOO solutions to different target model weights, enabling the target model to achieve the desired Pareto trade-off for the MOO problem.

Assume the target task maps from input $Y$ to output $Z$. We seek to minimize objective functions $f_1$ and $f_2$ having loss functions $\mathcal{L}_1$ and $\mathcal{L}_2$. Given correct output $Z^*$, we score $Z$ for each loss function $\mathcal{L}_i(Z, Z^*)$. A target model for this task $C_\Theta : Y \to Z$ with parameters $\Theta$ will yield loss $\forall_i \mathcal{L}_i(C_\Theta(Y), Z^*)$. The MOO problem is to find Pareto optimal $\Theta^*$ for the $f_1 = \mathcal{L}_1$ *vs.* $f_2 = \mathcal{L}_2$ trade-off.

The objectives $\mathcal{L}_1(\Theta), \mathcal{L}_2(\Theta)$ for SUHNPF are continuous differentiable functions of $\Theta$. This enables SUNHPF's guided double gradient descent strategy to efficiently search the space of model target parameters $\Theta$, mapping each to resulting loss values $(\mathcal{L}_1, \mathcal{L}_2)$. Training data resulting from this search allows SUHNPF to learn an $\epsilon$-bounded approximation $\tilde{M}(\Theta^*)$ to the weak Pareto optimal manifold.

As in prior Pareto Front Learning (PFL) work [Navon et al., 2021, Lin et al., 2020], this enables rapid model personalization at run-time based on user preferences. The neural MOO $Loss_{classifier}$ is a weighted linear combination of the user-prescribed objectives $(\mathcal{L}_1, \mathcal{L}_2)$. The classifier loss hyper-parameter $\alpha$ (trade-off value) is computed as a post-processing step corresponding to Pareto optimal classifier weights $\Theta^*$ for rapid traversal of arbitrary $(\alpha, \Theta^*)$ solutions. See **Appendix A** for additional details of the setup of SUHNPF as a hypernetwork to optimize a target model.

## 7.1 EVALUATION ON MULTI-TASK LEARNING

We evaluate on the same MTL image classification problems as in Navon et al. [2021]. Given two underlying source datasets, MNIST [LeCun et al., 1998] and Fashion-MNIST [Xiao et al., 2017a], Navon et al. [2021] report on three MTL tasks: MultiMNIST [Sabour et al., 2017], Multi-Fashion, and Multi-Fashion + MNIST. In each case, two images are sampled from source datasets and overlaid, one at the top-left corner and one at the bottom-right, with each also shifted up to 4 pixels in each direction. The two competing tasks are to correctly classify each of the original images: Top-Left (Task 1 or $f_1$) and Bottom-Right (Task 2 or $f_2$). We use 120K training and $20k$ testing examples and directly apply existing single-task models, allocating $10\%$ of each training set for constructing validation sets, as used in Lin et al. [2019]. Navon et al. [2021] found that PHN-EPO (henceforth PHN) was more accurate than other methods they compared, so we use PHN as our baseline.

We adopt the LeNet architecture [LeCun et al., 1998] as the target model to learn. Following prior MTL work [Sener and Koltun, 2018], we treat all layers other than the last as the shared representation function and put two fully-connected layers as task-specific functions. We use cross-entropy loss with softmax activation for both task-specific loss functions. Because cross-entropy loss functions are differentiable, we can use them directly as training objectives.

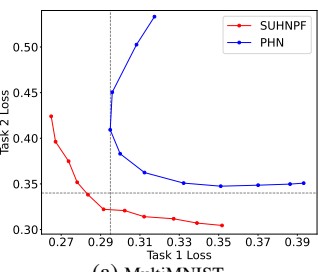 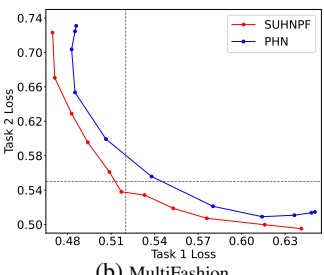 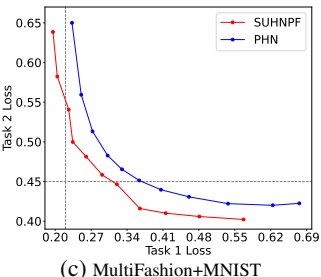

|                  |                  |                  |
|:----------------:|:----------------:|:----------------:|
| (a) MultiMNIST   | (b) MultiFashion | (c) MultiFashion+MNIST |

Figure 7: Cross-entropy loss on the test split for all three MTL datasets for SUHNPF *vs.* PHN. The 11 points shown for each method correspond (from left-to-right) to varying trade-offs preferences in minimizing the combined linear loss over objectives: $\alpha f_1 + (1 - \alpha)f_2$ for $\alpha \in \{1, 0.9, \ldots, 0\}$. The gray dashed-line show the best loss achieved by LeNet to classify a single image for each given task.

**Results.** We see SUHNPF *vs.* PHN results on dataset test splits in **Fig. 7**. Because SUHNPF defines a strict $\epsilon$-bound on error, we can assert its correctness on this basis alone. Visual inspection also shows that PHN returns dominated points (*e.g.,* top of MultiMNIST plot), whereas a Pareto front by definition includes only non-dominated points. Nonetheless, we cannot directly measure error *vs.* a known Pareto front because real MOO problems lack a simple analytical solution like synthetic benchmark problems. Of course, we can still compare relative performance of methods. We see that *SUHNPF achieves strictly lower loss than PHN across all user trade-off settings of $\alpha$ on all three datasets.*

Since the minimum loss $\min(f_1){=}\min(f_2){=}0$, for both objectives, the ideal point [Marler and Arora, 2004] for joint minimization is $(0, 0)$. A simple error measure for each point found is thus its $L2$ distance from $(0,0)$: $\sqrt{f_1^2 + f_2^2}$. **Table 5** reports this distance for each Pareto point found at each $\alpha$ (across methods and datasets). We also report the average over the 11 settings of $\alpha$. Overall, Table 5 quantifies what Fig. 7 depicts visually: SUHNPF performs strictly better for every Pareto point and thus also on average.

Table 5: SUHNPF *vs.* PHN on MTL tasks, measured by distance of each Pareto point found *vs.* the ideal loss point $(f_1, f_2) = (0, 0)$.

| Method | 0.0 | 0.1 | 0.2 | 0.3 | 0.4 | 0.5 | 0.6 | 0.7 | 0.8 | 0.9 | 1.0 | Avg |
|---|---|---|---|---|---|---|---|---|---|---|---|---|
| | | | | | | Trade-off values $\alpha$ | | | | | | |
| | | | | | | MultiMNIST | | | | | | |
| PHN | .621 | .585 | .539 | .504 | .486 | .478 | .483 | .494 | .508 | .521 | .527 | **.522** |
| SUHNPF | .500 | .478 | .464 | .448 | .441 | .434 | .441 | .443 | .452 | .457 | .465 | **.456** |
| | | | | | | MultiFashion | | | | | | |
| PHN | .877 | .872 | .853 | .813 | .784 | .773 | .779 | .797 | .816 | .826 | .829 | **.819** |
| SUHNPF | .862 | .819 | .792 | .773 | .757 | .746 | .754 | .758 | .767 | .793 | .810 | **.784** |
| | | | | | | MultiFashion+MNIST | | | | | | |
| PHN | .690 | .613 | .581 | .569 | .571 | .579 | .598 | .631 | .682 | .752 | .797 | **.642** |
| SUHNPF | .667 | .617 | .586 | .552 | .547 | .543 | .549 | .553 | .583 | .629 | .695 | **.593** |

## 8 UNDERSTANDING SUHNPF *VS.* PHN

While both SUHNPF and PHN are manifold-based, they differ in the type of manifold being learned. SUHNPF explicity maintains point sets $\mathcal{P}0$ and $\mathcal{P}1$ to learn the classification boundary between Pareto *vs.* non-Pareto points as per the FJC. PHN fits a regression surface over the set of points returned by LS or EPO. Since neither LS nor EPO are guaranteed to operate under non-convex settings (Section 3), those drawbacks are in turn inherited by PHN in using them. **Table 6** highlights the key differences. The distinction between a diffusive full-rank indicator *vs.* a low-rank regressor is further discussed in **Appendix B**.

Table 6: SUHNPF *vs.* PHN for Pareto front learning.

| Criteria | SUHNPF | PHN |
|---|---|---|
| Handle non-convexity | ✓ | ✗ |
| Supports constraints | ✓ | ✗ |
| Manifold Extractor | ✓ | ✓ |
| Nature of manifold | Diffusive full-rank indicator | Low-rank regressor |
| Optimality Criteria | Fritz-John Conditions | EPO solver |

## 9 CONCLUSION

Multi-objective optimization problems require balancing competing objectives, often under constraints. In this work, we described a novel method for *Pareto-front learning* (inducing the full Pareto manifold at train-time so users can pick any desired optimal trade-off point at run-time). Our SUHNPF Pareto solver is robust against non-convexity, with error bounded by a user-specified tolerance. Our key innovation over prior work's HNPF [Singh et al., 2021] is to exploit Fritz-John Conditions for a novel guided *double gradient descent* strategy. The scaling property imparts significant improvement in memory and run-time *vs.* prior OR and Multi-Task Learning (MTL) approaches. Results across synthetic benchmarks and MTL problems in image classification show clear, consistent advantages of SUHNPF in capability (handling non-convexity and constraints), denser coverage and higher accuracy in recovering the true Pareto front, and efficiency (time and space). Beyond empirical results, our conceptual framing and review of prior work also further bridges disparate lines of OR and MTL research.

Both SUHNPF and MTL methods assume differentiable evaluation metrics as training loss so optima can be found through gradient descent. However, loss can be a non-differentiable, probabilistic measure, such as in fairness-related tasks [Sacharidis, 2019, Valdivia et al., 2021]. This creates a risk of metric divergence between training loss *vs.* the evaluation measure of interest [Abou-Moustafa and Ferrie, 2012]. Continuing development of differentiable measures can help to address this [Swezey et al., 2021].

**Acknowledgments**. We thank the reviewers for their valuable feedback. This research was supported in part by Wipro and by Good Systems[2], a UT Austin Grand Challenge to develop responsible AI technologies. The statements made herein are solely the opinions of the authors.

---

[2] http://goodsystems.utexas.edu/

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
