# OpenReview forum: "Learning a Neural Pareto Manifold Extractor with Constraints"
_auai.org/UAI/2022/Conference — UAI 2022 Poster_

### Official Review · Reviewer_NEiQ · 2022-04-08

**Q2(1) Originality/Novelty:** 3
**Q2(2) Significance/Impact:** 3
**Q2(3) Correctness/Technical Quality:** 3
**Q2(6) Clarity Of Writing:** 4
**Q6 Overall Score:** 7
**Q8 Confidence In Your Score:** 4

**Q1 Summary And Contributions:**

This paper proposes a new method for multi-objective optimisation with continuous solution spaces and constraints. The methods are limited to problems with known and (once) differentiable objective and constraint functions. The method (SUHNPF) is evaluated on synthetic and multi-task classification problems, and a clear benefit is demonstrated over the state of the art.

**Q2 Assessment Of The Paper:**

More detailed information regarding each of these aspects is given below:

**Q2(4) Quality Of Experiments (Optional):**

3: Good: The experimental evaluation is adequate, and the results convincingly support the main claims.

**Q2(5) Reproducibility:**

3: Good: Key resources (e.g., proofs, code, data) are available and key details (e.g., proofs, experimental setup) are sufficiently well-described for competent researchers to confidently reproduce the main results.

**Q3 Main Strengths:**

The paper contains a clear mathematical analysis of why the method works, the memory complexity, and it contains extensive testing (albeit with 2-objective problems only).

**Q4 Main Weakness:**

I do not believe the paper has any weaknesses that aren't easily fixed but there are a few omissions that I think should be addressed:

- the paper often claims that SUHNPF has a better spread than other algorithms, however this is never quantified, and the figures do not always make this abundantly clear in my opinion.

- runtime complexity seems to be hinted at in the intro, but is not provided.

- there are no error margins provided in the results

- while promised, appendix G does not provide and analysis and justification for the LS results

- FJC are described as necessary, but never as sufficient. I assume this is an omission as later in the paper it claims that SUHNPF is guaranteed to output only weak Pareto-undominated solutions.

**Q5 Detailed Comments To The Authors:**

The following are all isolated comments, ordered by priority (in my opinion)

- One example of better scalability (table 3) doesn't really convey better scalability. It would be better to vary the number of input dimensions. Furthermore, the claim is that it would scale better runtime-wise, but the runtimes aren't provided in table 3. (I believe they will be worse, but I'd prefer to see them.)

- The scalability in the number of objectives is not addressed at all

- The realistic example (multi-task learning) has a convex user utility (weighted linear combination) of the objectives. As such a full Pareto-front learning method is not necessary for this task at all, and especially because it's only two objectives, more efficient methods are likely to exist that have better guarantees in terms of user utility. (For example, optimistic linear support, or, recursive simplex subdivision).

- The motivation for why Pareto-front methods are necessary is not complete. Firstly, the user utility should be non-convex (a condition which, as illustrated by my previous point, is not fulfilled by the MTL example). Furthermore, a mixture policy (i.e., taking multiple solutions and randomising over them), should not be allowed (or more precisely, the utility should not be over the expected fitness vector over this stochastic mixture over multiple solutions). Please see: Vamplew, Peter, et al. "Constructing stochastic mixture policies for episodic multiobjective reinforcement learning tasks." Australasian joint conference on artificial intelligence. Springer, Berlin, Heidelberg, 2009. In other words, simply because a problem has deterministic outcomes, does not mean that the solution can't be randomised.

- many references are to ArXiv versions - please update them to the full conference/journal/workshop versions. (In my experience, it is not possible to trust the results presented only in non-peer reviewed venues.)

- HNPF, SUNHPF abbreviations are used before being defined. alpha is never formally defined.

===== In response to the rebuttal =====

Firstly, let me state that I am still in favour of accepting this paper. However, there are a few points in the rebuttal I disagree with.
1) a mixture policy on the faces of the convex hull that Pareto-dominates any policy on a non-convex region of the Pareto front is guaranteed to exist. One does not actually mix the networks, only stochastically selects which solution to execute, leading to a Pareto-dominating /expected value/ for the mixture policy. It is this stochastic (and if you want to be technical, also non-stationary) execution which needs to be disavowed for motivating a Pareto-front. (Please also note that your characterisation of Vamplew et al.'s (2009) methods is just plain wrong: one would only mix solutions that together form one of the faces of the convex hull.)
2) That neural networks lead to non-convex optimisation problems in terms of the parameter-space of the neural networks whether to optimise for a single objective, a linear combination of objectives, or any Pareto-optimal solution, is very true indeed. That does however not take away the fact that optimising for a single objective is easier than for a linear combination of objectives (with unknown weights), and that is in turn easier than optimising for any Pareto optimal solution. For example, for very hard problems indeed, e.g., a partially observable Markov decision process - which is undecidable - one can still get arbitrary (epsilon) close to optimal for a single-objective instance, and for a linear combination of objectives, but this has not been shown for non-convex combinations. Even in the case of neural networks, there is research where linear preference weights are injected into a neural network after the representation layers, leveraging the linear weights property computationally.

The point is: assuming Pareto optimality leads to a sufficient, but not always a necessary solution set. Taking a bigger solution set might seem safer, but it leads to a harder optimisation problem, and one might be wasting a lot of computation time (and for some problems, even a shot at a solution set with better guarantees). Therefore, assuming the Pareto front as the solution set needs to be properly motivated.

That being said, I believe this paper has a lot of merits, and I will vote to accept it, but please consider providing a more thorough motivation.



**Q7 Justification For Your Score:**

Nice paper, I learned something new from it, and I think it can have an impact on other subfields of AI as well, such as multi-objective decision-theoretic planning.

**Q9 Complying With Reviewing Instructions:**

1: Yes.

---

### Official Review · Reviewer_iV6p · 2022-04-14

**Q2(1) Originality/Novelty:** 2
**Q2(2) Significance/Impact:** 2
**Q2(3) Correctness/Technical Quality:** 3
**Q2(6) Clarity Of Writing:** 3
**Q6 Overall Score:** 5
**Q8 Confidence In Your Score:** 4

**Q1 Summary And Contributions:**

This paper aim at balancing competing objectives in dealing with multi-objective optimization problems. It is an improvement of HNPF in terms of scalability. It exploits Fritz-John Conditions and proposed a double gradient descent strategy. Experimental results show superiority in multi-objective optimization and multi-task learning.

**Q10 Ethical Concerns (Optional):**

NAN

**Q2 Assessment Of The Paper:**

More detailed information regarding each of these aspects is given below:

**Q2(5) Reproducibility:**

2: Fair: Key resources (e.g., proofs, code, data) are unavailable but key details (e.g., proof sketches, experimental setup) are sufficiently well-described for an expert to confidently reproduce the main results.

**Q3 Main Strengths:**

1. The use of Fritz-John Conditions is novel to my knowledge.

2. The figures are clearly plotted, and show the advantages of their proposed method.

**Q4 Main Weakness:**

1. The authors claim that their proposed method aims at 'inducing the full Pareto manifold at train-time so users can pick any desired optimal trade-off point at run-time', however, this is not well illustrated in the experiment part. And the motivation for doing this should be made clearer.

2. I'm not sure whether the 'epsilon-bounded approximation' is demonstrated clearly in the paper.

3. Table 5 uses a specific combination of f1 and f2, however, this is against the purpose of multi-objective optimization.

**Q5 Detailed Comments To The Authors:**

- The author mentioned in Section 4 that Fritz-John Condition is a necessary condition. I'm curious about if it is not necessary and sufficient, can we use it that way?

- Some part of the writing is hard to understand. For example, 1.'The classifier loss hyper-parameter \alpha (trade-off value) is computed as a postprocessing step corresponding to Pareto optimal classifier weights for rapid traversal of arbitrary solutions.' makes me confused about the computing of \alpha. 2.'we treat all layers other than the last as the shared representation function and put two fully-connected layers as task-specific functions' I wonder whether there is one last layer or two layers? Whether the tasks share the two layers or not?

- Other problematic sentences include 1.'With HNPF, Pareto points for use as training data data'. 2.'Both SUHNPF and MTL methods assume differentiable evaluation metrics as training loss so optima to be found through gradient descent.' I guess it should be 'so that optima can be found'.

**Q7 Justification For Your Score:**

The use of Fritz-John Conditions is novel to my knowledge. [major strengths]

The authors claim that their proposed method aims at 'inducing the full Pareto manifold at train-time so users can pick any desired optimal trade-off point at run-time', however, this is not well illustrated in the experiment part. And the motivation for doing this should be made clearer. [major weakness]



**Q9 Complying With Reviewing Instructions:**

1: Yes.

---

### Official Review · Reviewer_UzJn · 2022-04-15

**Q2(1) Originality/Novelty:** 3
**Q2(2) Significance/Impact:** 2
**Q2(3) Correctness/Technical Quality:** 3
**Q2(6) Clarity Of Writing:** 3
**Q6 Overall Score:** 6
**Q8 Confidence In Your Score:** 2

**Q1 Summary And Contributions:**

In this paper, the authors propose the method for MOO. Specifically, they propose a double gradient descent strategy with Fritz-John Conditions. The experiments demonstrate the effectiveness of the proposed method.

**Q10 Ethical Concerns (Optional):**

No.

**Q2 Assessment Of The Paper:**

More detailed information regarding each of these aspects is given below:

**Q2(4) Quality Of Experiments (Optional):**

2: Fair: The experimental evaluation is weak: important baselines are missing, or the results do not adequately support the main claims.

**Q2(5) Reproducibility:**

3: Good: Key resources (e.g., proofs, code, data) are available and key details (e.g., proofs, experimental setup) are sufficiently well-described for competent researchers to confidently reproduce the main results.

**Q3 Main Strengths:**

The introduction of Fritz-John Conditions is nice.
The paper is well organized and provides some necessary analysis.
There are sufficient experiments to support their method.



**Q4 Main Weakness:**

I think the current version lacks a clear illustration of their motivation.



**Q5 Detailed Comments To The Authors:**

I suggest the authors use a unified format in Reference. For example, there are page numbers for "Multi-task learning with user preferences: Gradient descent with controlled ascent in pareto optimization", but are not for "Minimax pareto fairness: A multi objective perspective".

It will be better if the authors could provide an experiment on the efficiency.


**Q7 Justification For Your Score:**

From my pointview, the proposed double gradient descent strategy is novel. In addition, the paper is clearly written. Hence I give a positive score.

**Q9 Complying With Reviewing Instructions:**

1: Yes.

---

### Decision · Program_Chairs · 2022-05-15

**Decision:**

Accept (Poster)

**Comment:**

Meta Review: The reviewers reach a consensus on the acceptance. The authors are encouraged to take all the comments into consideration and further improve the paper in the camera ready.